# Spatial and temporal evolution of ecotourism development level and its driving factors under the perspective of sustainable development: the case of Ili river valley

**Pengkai Zhao**[ID]°, **Haojie Sun**[ID]*, **Jiangling Hu**‡, **Xinyu Zhao**‡, **Changying Song**‡, **Xueting Xu**‡

College of Geographical Science and Tourism, Xinjiang Normal University, Urumqi, China

☉ These author equally contributed to this work.
‡ JH, XZ, CS and XX also equally contributed to this work.
* 107622001010082@xjnu.edu.cn

## Abstract

Ecotourism, as an ideal model for sustainable tourism development, is a response to ecological problems and the way tourism is developed. The foundational elements of ecotourism serve as the basis for the development level evaluation index system. Ten counties and cities in the Ili River Valley are evaluated for their level of ecotourism development between 2010 and 2019 using the entropy weight TOPSIS approach. Using the standard deviation ellipse, classic Markov chain, and spatial Markov chain, the temporal and spatial evolution characteristics are examined. Geographic detectors are utilized to explore the driving factors of ecotourism development. The data indicates that: (1) With a notably diverse spatial structure, the growth rate of ecotourism varies among the counties and cities in the Ili River Valley. (2) The level of comprehensive ecotourism development is continuously improving, with significant gradient differences in spatial distribution, forming a dynamic spatial pattern of 'high in the north and low in the south.' (3) The standard deviation ellipses of each year show a "northwest – southeast" direction, and basically form a stable migration rule from northwest to southeast; (4) The level of tourism income and economic development have a significant impact on the development of ecotourism, and the influence of tourism reception capacity and industrial structure level is gradually enhanced, while the promotion effect of ecological environment level is not significant. The interaction of the two factors is greater than that of the single factor, indicating that the interaction connection is facilitated by the two elements. The findings of the study can offer some theoretical underpinnings and scientific references for raising the degree of ecotourism development and encouraging the Ili River Valley's tourism industry's sustainable growth.

**Data availability statement:** All relevant data are within the article and its Supporting Information files.

**Funding:** This research was supported by the Xinjiang Uygur Autonomous Region Social Science Foundation Project Self-help (Project No. 2023BYJ033).

**Competing interests:** The authors have declared that no competing interests exist.

## 1. Introduction

Ecotourism is a type of tourism that takes place in natural areas and emphasizes the law of minimum environmental impact. It is an effective way to realize the sustainable development of modern tourism [1]. Its goal is to maintain sustainable management of the ecological environment [2] by emphasizing the concepts of environmental education and tourist protection, as well as respect for local cultures and the promotion of local development. In 1998, ecotourism as a core concept of environmental development strategy was first formally proposed. The World Conference on Sustainable Tourism Development, which took place in the Canary Islands, Spain, in April 1995, highlighted the interdependence of tourism and the environment and made it abundantly evident that the sustainable growth of tourism could only be achieved by closely integrating tourism and environmental protection. The historic "Charter for Sustainable Tourism Development" and "Action Plan for Sustainable Tourism Development" were also adopted during the conference [3], offering crucial framework and direction for the long-term growth of international travel. As a world-class tourism resource-rich area, the Ili River Valley is blessed with a unique ecological environment, with almost all the rich natural landscapes except the ocean and high-quality eco-tourism resources such as the Tianshan Mountain World Natural Heritage Site, China's most beautiful mountain grasslands, and China's famous Chinese Heavenly Horses. However, the development of ecotourism faces many challenges, such as monopolization by external enterprises to the exclusion of local farmers and herdsmen, shortsighted development by enterprises to destroy tourism resources, difficulty to balance infrastructure and ecological environment, weak market awareness and industrial development concepts of local residents, limited participation of farmers and herdsmen, and the lack of effective competition mechanisms. At the same time, ecotourism must grow sustainably in order to meet the demands of sustainable development and environmental preservation, preserve the natural ecosystem while judiciously using tourism resources, create a situation where both the economy and the environment benefit, and prevent the "tragedy of the commons".

## 2. Literature review

Ecotourism, as an important form of alternative tourism and sustainable tourism practice [4], has been attracting the attention of scholars at home and abroad since its inception. In terms of research content, European and American scholars, due to their early start, have transitioned from studies related to the connotation of ecotourism [5–7], stakeholders [8–10], and source markets [11–12] to ecotourism's eco-tourism sites [13–14], impacts [15–16], and assessment and protection [17–18]; while Central Asian scholars have combined with the actuality of China's eco-tourism resources, from ecotourism theory [19], method [20] research to sustainable development [21–23], value assessment [24–25] and ecological protection [26] research [27]. As for the research methodology, available studies have also experienced the developmental process of shifting from qualitative research, such as descriptive analysis [28–30] and interview method [31–34], to quantitative research, such as model analysis [35–36] and statistical analysis [37–39]. After that, the academic community began to focus on the important topic of ecotourism development level and deeply explored the synergistic relationship between ecotourism development level and regional development [40], regional economy [41], as well as the distribution characteristics of the development level in the spatial pattern [42]. Among the representative research results, In order to evaluate the degree of ecotourism sustainability in four distinct categories of protected areas in Iran, Parvaneh [43] et al. created and retrieved sustainable ecotourism indicators from the three aspects of natural-environmental, demographic-social, and economic-institutional dimensions; To investigate

the sustainable growth of ecotourism in national parks, Li [44]et al. developed a three-way game evolution model called "local government-tourism enterprises-tourists," with an emphasis on coordinating the interests of various stakeholders; In-depth analysis of the temporal and spatial differentiation phenomenon of the coupled and coordinated development of ecology, tourism, and culture in the Yellow River basin was conducted by Zhang Zhongwu [45] et al. using a range of research techniques, including the entropy method, the coupled coordination degree model, and the geoprobe model.

In conclusion, even though the existing research has established a strong basis for this subject, there are still a lot of things that could be done better. The research content lacks depth and breadth, and most studies are still in the primary stage of theoretical discussion. There is insufficient research on the heterogeneity of ecotourism development levels in different regions, as well as research exploring its spatial and temporal evolution and influencing factors over a long period of time. Most research on ecotourism focuses on smaller scales, such as parks, scenic spots, or individual cities, or larger scales, such as provinces and countries. However, there is relatively little attention paid to the development of ecotourism in underdeveloped areas at the mesoscale. The ten county-level cities in the Ili River Valley serve as the primary focus of this paper's research. It uses the entropy weighted TOPSIS method to measure the development level of ecotourism in the Ili River Valley from 2001 to 2019 and then combines the spatial Markov chain to analyze its spatio-temporal dynamic evolution characteristics. Finally, it uses geographic probes to analyze the primary factors that drive the development level of ecotourism in the Ili River Valley in order to provide references to promote ecotourism in the region. We offer a reference base to support the sustainable growth and high caliber of ecotourism in the Ili River Valley.

## 3. Overview of the study area

The Ili River Valley is located in the Xinjiang Uygur Autonomous Region's Ili Kazakh Autonomous Prefecture (80°09′ ~ 84°56′W, 42°14′ ~ 44°53′N). It is encircled by mountains on three sides and is situated in the western portion of the Tien Shan Mountains. It is part of the temperate continental semi-arid climate, which is pleasant and humid. In the arid region of the Asian-European continent, this region can be regarded as a wet island. The region is known as a 'wet island' in the arid zone of the Asian-European continent due to its fertile land, abundant water sources, vast grasslands, and abundant produce. It is often referred to as an 'oasis in Central Asia', as shown in Fig 1. The Ili River Valley comprises eight counties and two cities, namely Yining City, Huoerguosi City, Yining County, Nileke County, Xinyuan County, Gongliu County, Tekesi County, Zhaosu County, Chabuchaerxibo Autonomous County, and Huocheng County [46]. At its broadest point, it is around 275 km from north to south and 360 km from east to west. The area covers 56,400 square kilometers and exhibits a clear difference in the spectrum of vertical climatic zones. It boasts a rich and colorful array of ecotourism and climatic resources, including snowcapped mountains, forests, grasslands, lakes, and other types of landforms. As a key strategic pillar industry in the Ili River Valley, tourism is increasingly driving Ili's superior social and economic growth. 38,293,700 tourists had visited the Ili River Valley as of December 2022, bringing in 20.12 billion yuan in tourism earnings, or 16.8% of the region's GDP.

## 4. Methods of research and sources of data

### 4.1 Research methods

**4.1.1 Indicator system construction.** This study develops an assessment index system of ecotourism development level with 20 indicators in four dimensions based on the

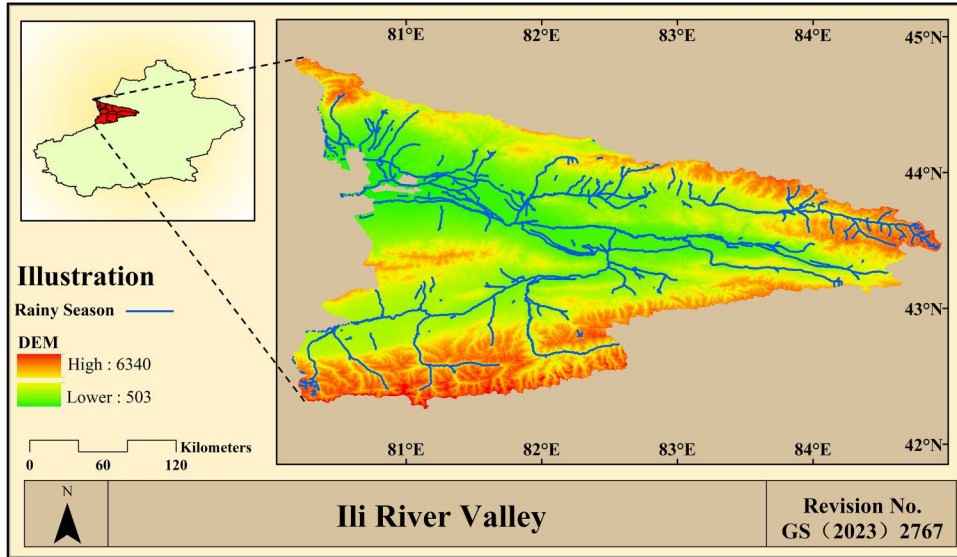

**Fig 1. Overview map of the Ili River Valley.** This map is based on the standard map with review number Xin S(2022)021 downloaded from the geographic information service platform of the Xinjiang Uygur Autonomous Region Department of Natural Resources, with no modifications to the base map(Same as below).

fundamentals of sustainable ecotourism development: ecotourism environment, ecotourism economy, ecotourism society, and ecotourism sustainable development potential, appropriate reference to the current pertinent research findings, while adhering to the criteria of systematization, scientificity, and data availability (Table 1).

Ecotourism focuses on protecting ecosystems and utilizing resources sustainably. It also considers economic benefits and employment opportunities, social impacts and cultural inheritance, and assesses future development potential and risks for sustainable development[42,47–49]. The evaluation of these four dimensions can provide a thorough picture of the state of ecotourism growth as well as its future prospects. This facilitates the development of ecotourism in harmony with society, the economy, and the environment and offers a scientific foundation and guidelines for advancing the industry's sustainable growth.

## 4.2 Entropy weight TOPSIS method

Ten counties and cities in the study area were evaluated for their level of ecotourism development using the entropy weight TOPSIS approach. An improved version of the conventional TOPSIS evaluation approach is the entropy weight TOPSIS method. The weights of the evaluation indexes are determined using the entropy weight approach, and the TOPSIS approach with the technique of positive-negative ideal solution is then used to rank the relative proximity of the evaluation objects [42]. The main steps are as follows:

(1) Matrix normalization is used to remove the impact of various indicator scales. The normalized matrix $X$, given $n$ evaluation items and $m$ evaluation indicators, is as follows:

$$X = \begin{bmatrix} x_{11} & x_{12} & \cdots & x_{1m} \\ x_{21} & x_{22} & \cdots & x_{2m} \\ \vdots & \vdots & \ddots & \vdots \\ x_{n1} & x_{n2} & \cdots & x_{nm} \end{bmatrix}$$

**Table 1. Evaluation index system of ecotourism development level.**

| Standardized Layer | Indicator Layer | Unit | Nature |
|---|---|---|---|
| Ecotourism Environment | NDVI (normalized vegetation index) | – | Forward |
| | Waste gas emissions | Million Nm³ | Negative |
| | Wastewater emissions | ten thousand t | Negative |
| | Smoke and dust emissions | t | Negative |
| | Comprehensive utilization rate of solid waste | % | Forward |
| Ecotourism Economy | Total tourism revenue as a share of GDP | % | Forward |
| | Number of tourists received | ten thousand people | Forward |
| | Employment in tertiary industry | ten thousand people | Forward |
| | Investment in fixed assets of the whole society | 10K CNY | Forward |
| Ecotourism Site Society | Population at the end of the year | ten thousand people | Forward |
| | Urbanization rate | yuan | Forward |
| | Fiscal Expenditure | yuan | Forward |
| | GDP per capita | % | Forward |
| | Night Light | – | Forward |
| | Natural population growth rate | ‰ | Forward |
| **Standardized Layer** | **Indicator Layer** | **Unit** | **Nature** |
| Ecotourism Sustainability Potential | Gross Regional Product | 10K CNY | Forward |
| | Number of A and scenic spots | pcs | Forward |
| | Number of guest rooms in star-rated hotels | pcs | Forward |
| | Number of Students in Secondary Schools | people | Forward |
| | Number of sanitary beds | sheet | Forward |

For its normalized matrix $Z$, each element in $Z$ is normalized by $Eq$:

$$z_{ij} = x_{ij} / \sqrt{\sum_{i=1}^{n} x_{ij}^2}$$

(1)

Every element in a column is divided by the square root of the total squares of all the elements in that column.

(2) Determine the $i$th program's value under the $j$th indication as a percentage of the indicator $p_{ij}$. At this time, the normalizing matrix $Z$ is:

$$Z = \begin{bmatrix} z_{11} & z_{12} & \cdots & z_{1m} \\ z_{21} & z_{22} & \cdots & z_{2m} \\ \vdots & \vdots & \ddots & \vdots \\ z_{n1} & z_{n2} & \cdots & z_{nm} \end{bmatrix}$$

Following

$$P_{ij} = z_{ij} / \sum_{i=1}^{n} z_{ij}, \left( j = 1, 2, \cdots, m \right)$$

(2)

Calculate the entropy value of the $j$th indicator $E_j$ when $P_{ij} = 0$, $P_{ij} ln P_{ij} = 0$

$$E_j = -\frac{1}{\ln n} \sum_{i=1}^{n} P_{ij} \ln P_{ij}, \left( j = 1, 2, \cdots, m \right)$$

(3)

(4) Calculate the coefficient of variation $G_j$ and weight $W_j$ for the $j$th indicator:

$$G_j = 1 - E_j \tag{4}$$

$$W_j = \frac{G_j}{\sum_{j=1}^{m} G_j} \tag{5}$$

(5) Identify the optimal positive and negative solutions. The negative ideal solution is the least value of each column element, while the positive ideal solution is the greatest value of each column element. At this time, the normalized matrix $Z$ is:

$$Z = \begin{bmatrix} z_{11} & z_{12} & \cdots & z_{1m} \\ z_{21} & z_{22} & \cdots & z_{2m} \\ \vdots & \vdots & \ddots & \vdots \\ z_{n1} & z_{n2} & \cdots & z_{nm} \end{bmatrix}$$

(6) Construct the weighting matrix and multiply each column of the normalized matrix $Z$ with the corresponding weight:

$$Z = \begin{bmatrix} \omega_1 * z_{11} & \omega_2 * z_{12} & \cdots & \omega_m * z_{1m} \\ \omega_1 * z_{21} & \omega_2 * z_{22} & \cdots & \omega_m * z_{2m} \\ \vdots & \vdots & \ddots & \vdots \\ \omega_1 * z_{n1} & \omega_2 * z_{n2} & \cdots & \omega_m * z_{nm} \end{bmatrix}$$

(7) Utilizing the negative ideal solution distance, determine the relative proximity $S_i$ between the assessed object and the optimal solution.

Unnormalized relative proximity:

$$S_i = \frac{D_i^-}{D_i^+ + D_i^-} \tag{6}$$

The relative proximity of the normalised:

$$S_i = S_i / \sum_{i}^{n} S_i \tag{7}$$

$S_i$ takes values between 0 and 1. The higher the score, the higher the level of ecotourism development.

## 4.3 Markov chain

### 4.3.1 Traditional Markov chain.
The Markov chain is primarily used to depict the probability distribution of a socio-economic event in an area that progressively shifts from one state to another by creating a state transfer probability matrix. It is capable of efficiently analyzing the dynamic shifts in the Ili River Valley's ecotourism development level over time. Continuous data is discretized using this method into $k$ categories. A transfer probability matrix

is created by the transfer between different categories in different years, and the maximum likelihood estimate approach is used to calculate the probability of category transfer [50].

$$P_{ij} = \frac{n_{ij}}{n_i} \tag{8}$$

$P_{ij}$ is a probability that, over the course of the research period, a city will change from category $i$ to type $j$ in the year after $(t + 1)$. During the study period, the total number of counties and cities that changed from type $i$ to level $j$ ecotourism development in year $t + 1$ is represented by the variable $n_{ij}$. Last but not least, $n_i$ is the total number of cities that fall under type $i$ over the course of the study.

**4.3.2 Spatial Markov chain.** Researchers have discovered that a region's growth and evolution are significantly influenced by its geographic proximity [51]. A conditional notion of "spatial lag" is presented and classified into $k$ categories according to the classical Markov chain's transfer probability matrix, taking into account the spatial features of local occurrences. It is possible to partition the $k * k$ transfer probability matrix into $k$ $k * k$ transfer conditional probability matrices.

## 4.4 Geo-detectors

Geo-detectors are powerful tools for detecting the spatial dissimilarity of geographic phenomena and their influencing factors [52]. In order to determine the influencing variables and their interactions with the growth of ecotourism in the Ili River Valley, this paper employs divergence and factor identification as well as interaction detection. The primary analysis done by factor identification and divergence is how much the various contributing factors account for the degree of ecotourism growth in the city.

$$q = 1 - \frac{1}{N\sigma^2} \sum_{h=1}^{L} N_h \sigma_h^2 \tag{9}$$

The formula's $q$ value, whose value ranges from 0 to 1, represents the degree to which the driving factor explains the level of ecotourism development. The factor's influence on the expansion of ecotourism increases with a higher $q$ value. $N$ and $\sigma^2$ are the number of sample units and the variance of the whole region, respectively; $N_h$ and $\sigma_h^2$ are the sample size and the variance of the influencing factors of the hth category, respectively; $L$ is the number of classifications for the influencing factors of the $h$th category.

## 4.5 Data sources

Due to the substantial impact of COVID-19 on tourism, the research period for this work is 2010–2019, and the main data sources are statistical data and basic geographic information data. The socio-economic information among them is primarily derived from statistical data released by the local government; this includes the China Ili Kazakh Autonomous Prefecture Statistical Yearbook from 2010 to 2019. Some of the missing information can be found in each county's and city's annual statistical bulletin. The bulletin of environmental conditions of counties and cities additionally offers certain ecological and environmental protection data in addition to the previously mentioned figures. Additionally, the NVDI data for this study was provided by the Resource and Environment Science and Data Center of the Chinese Academy of Sciences (https://www.resdc.cn/), while the night lights data was obtained from the National Science and Technology Basic Condition Platform-National Earth System Science Data Center

(https://www.geodata.cn/main/#/). For some of them, linear interpolation was used uniformly to fill in the missing values.

## 5. Results and Analysis

### 5.1 Examination of the Ili River Valley's ecotourism development level's spatiotemporal pattern

**5.1.1 Spatio-temporal analysis of the development level of each dimension.** The degree of ecotourism development in the counties and cities in the Ili River Valley was assessed using the entropy weight TOPSIS approach. Using the ArcGIS10.8 natural breakpoint method, the time series variations of the development index of each ecotourism dimension in the Ili River Valley were shown (Fig. 2) and divided into five levels: low, lower, medium, higher, and high. On the chronological change (Fig 2), the development level of all dimensions showed an increasing and then decreasing trend, with the ecotourism environment having the highest volatility, the ecotourism sites' social sustainability and potential level of sustainable development being the second highest, and the ecotourism economy having the lowest volatility of sustainable development.

Overall, the state of ecotourism is excellent, but in 2015, there was a dip because of the protracted ecological recovery cycle and the sluggish impact of ecological benefits. Because Zhaosu County, Xinyuan County, Nileke County, and other counties and municipalities strengthened the protection of the ecological environment and saw short-term success with their solid waste management, the ecotourism environment development index reached a peak of growth in 2014. The social sustainable development index of ecotourism places, on the other hand, shows a fluctuating upward trend of first increasing and then decreasing, with the development index showing a slow upward trend in 2010-2013 and a slow downward trend in 2014-2018, with the risk of sustained decline. The Ili River Valley has a comparatively high demand for tourists as well as comprehensive social and economic security, which is favorable to the sustainable development of urban ecotourism according to the ecotourism economy's general growth trend and sustainable development potential.

In terms of spatial distribution, the ecotourism environment development index generally exhibits a pattern of being low in the east and high in the west (see Fig. 3). The high-level cities are situated on the periphery of the study area, far from the highly urbanized regional development centers. These cities boast higher levels of natural resilience, less human pollution, and a generally favorable ecological environment. However, Yining City, Yining County, and other urbanized areas have a lower level of ecological quality due to the increasing contradiction between economic development and environmental protection. The influx of population and urban construction have further exacerbated this issue, posing a threat to their ecological environment.

The northeast has a high concentration of ecotourism, while the southwest has a low concentration, according to the ecotourism economic development index. This index reflects the ability of tourist destinations to provide high-quality tourism products and hospitality services. To promote balanced development, the Ili River Valley should actively develop ecotourism resources, increase investment in the tourism industry, and avoid an unbalanced supply level. The distribution of ecotourism sites shows a significant difference between the north and south regions, with a higher concentration in the north and a lower concentration in the south. Compared to southern cities, northern cities have more ecotourism resources and a higher demand for ecotourism with a greater annual growth rate. Low and lower-level cities are more prevalent in the southern part of the study area, and the overall social development level of tourist sites is lower. Southern cities can leverage the development of northern cities to

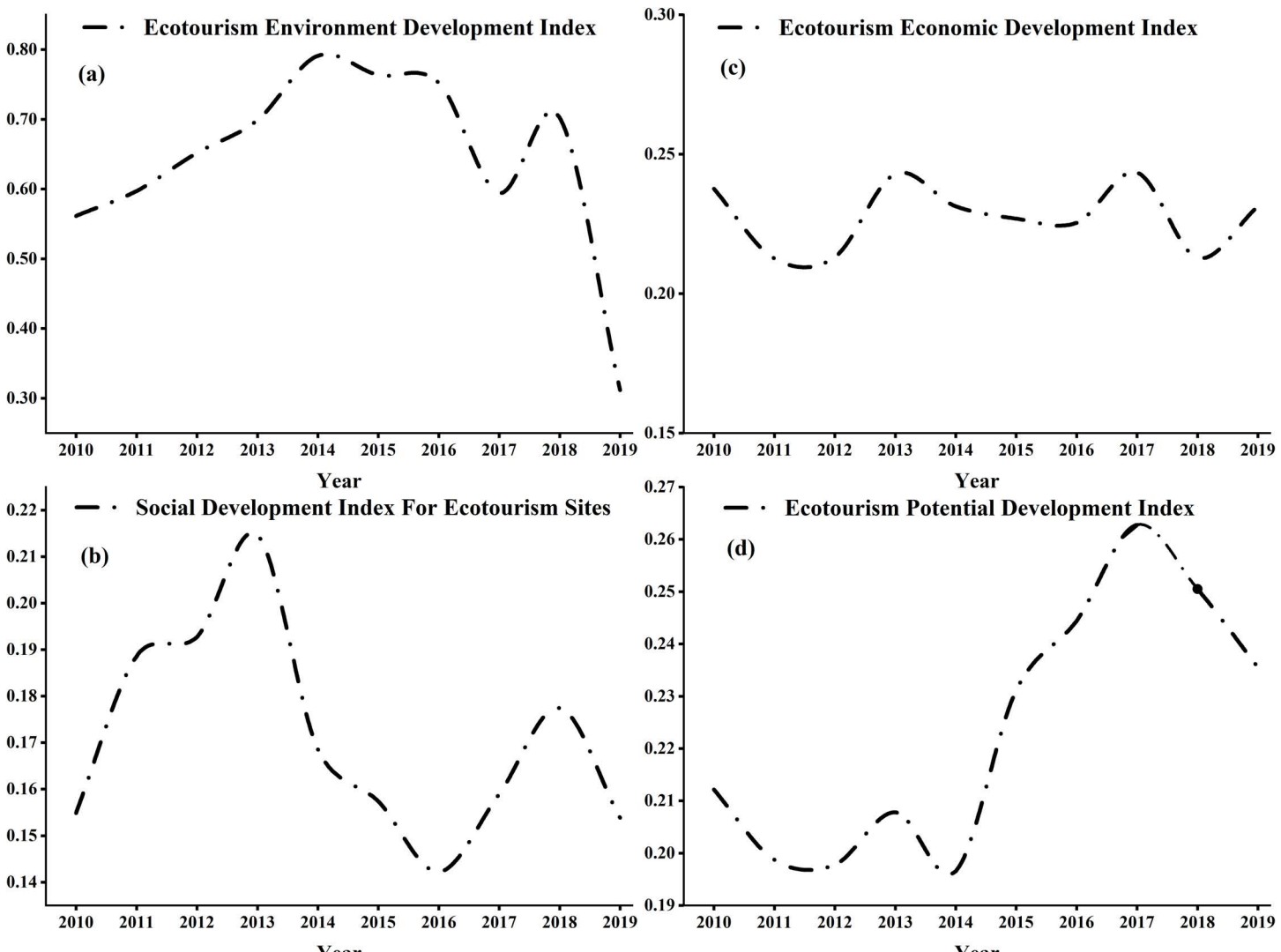

**Fig 2. Variations over time in the Ili River Valley's time series development index for every aspect of ecotourism.**

create unique ecotourism products, enhancing tourist appeal and promoting urban ecotourism's social development potential. The spatial distribution pattern of ecotourism's sustainable development potential shows a center as the depression, gradually increasing outward. High-value areas are distributed in strips across Yining City, Yining County, and Xinyuan County. Additionally, there are high-value areas located in Zhaosu County and Tekesi County to the south. The Ili River Valley can provide stronger economic and material support for ecotourism due to rapid economic development and improved transportation infrastructure.

**5.1.2 Analyzing integrated development levels' temporal and spatial trends.** Regarding changes over time, the Ili River Valley's ecotourism comprehensive development index increased from 0.2106 to 0.2494, a 3.88% rise. The index showed slight variations between years and an overall trend of slight increases and fluctuations, indicating a gradual and slow process towards raising the level of development of ecotourism in the region. While the number of cities at the medium level declined to varied degrees, the number of cities at the

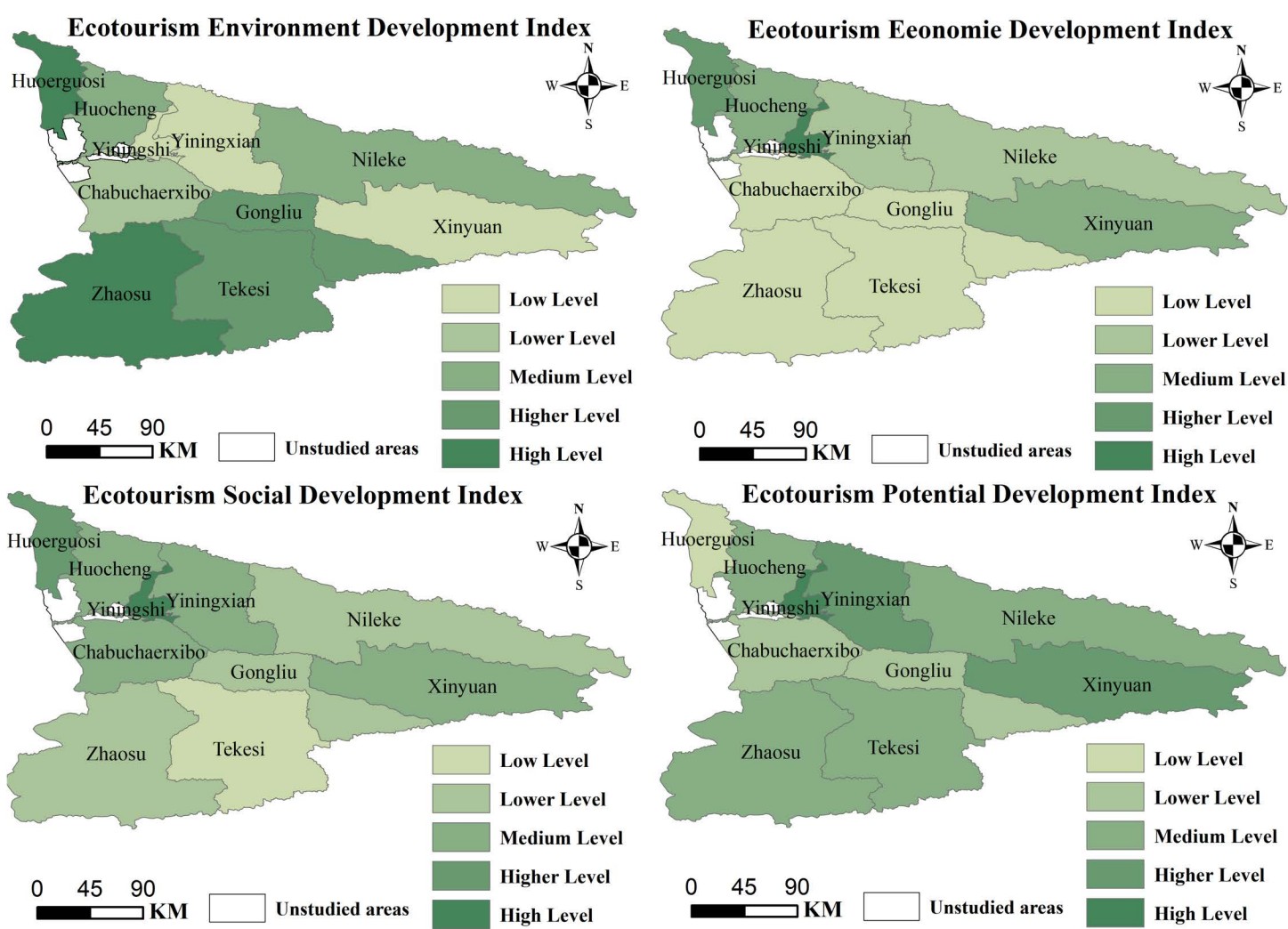

**Fig 3. The level of development of the dimensions of ecotourism in the Ili River Valley.** This map is based on the standard map with review number Xin S(2022)021 downloaded from the geographic information service platform of the Xinjiang Uygur Autonomous Region Department of Natural Resources, with no modifications to the base map(Same as below).

head and tail levels rose. Among them, Yining City is always in the first place, showing the development trend of "high level and fast growth"; Huoerguosi City and Xinyuan County have the second highest development index in all dimensions, with a high and stable development level; Yining County and Huocheng County have lower ecological environment indexes, but due to the high tourism economy, social development, and tourism potential, their comprehensive development level is always in the forefront; Zhaosu County, Nileke County, and Gongliu County show significant growth with the continuous highlighting of regional characteristics and culture, as well as favorable policy support. With the continuous highlighting of regional characteristics and culture, coupled with favorable policy support, Zhaosu County, Nileke County, and Gongliu County saw significant growth. During the same period, the development index of Tekesi County and Chabuchaerxibo Autonomous County declined significantly, gradually falling to the low gradient region. The main causes are the dearth of ecotourism resources, the ecological environment's sluggish recovery, which reduces the demand for ecotourism, and typical ecological cities, such as Zhaosu County, have

excellent ecological environments, but supply, demand, and security have become key factors restricting the enhancement of its comprehensive level, and thus its integrated development level is not high.

To sum up, high-level counties and cities have not yet formed a large-scale layout in the Ili River Valley, and medium and low-level cities are widely distributed. High-level and higher-level cities are primarily found in areas with good comprehensive conditions because the development of ecotourism depends not only on the benefits and drawbacks of the ecological environment but also on the combined effect of location conditions, economic development, policy support, and other factors. As a result, a single element cannot directly affect the overall development of ecotourism. As a result, high-level and higher-level cities are typically found in regions with exceptional overall conditions, but the "short board" effect frequently limits cities with lower levels of development.

From the standpoint of spatial evolution, the Ili River Valley's general level of ecotourism growth has essentially produced a dynamic geographical pattern of "high in the north and low in the south" (Figure 4). Lower-level cities in the southwest show a trend of contraction, while low-level counties and cities in the southeast show a spreading trend. The higher-level and high-level cities in the northwest of the spatial distribution characteristics of agglomeration show a more pronounced regional agglomeration effect; medium-level counties and cities are concentrated in the nearby higher-level cities, mainly due to the radiation function of the high-level cities, which encourages the transfer of ecotourism elements to the low-gradient cities and drives the growth of the surrounding counties and cities. With a development index that is constantly dropping and a distribution of low-level cities that shows the pattern of growing and then contracting, Chabuchaerxibo Autonomous County, Gongliu County, and Tekesi County are at the low level of development. In contrast, Yining and Huoerguosi cities have the highest level of comprehensive ecotourism development compared to other cities, and Yining and Huocheng counties have reached the medium level or above. The aforementioned demonstrate that the degree of ecotourism growth in the Ili River Valley has notable socioeconomic preferences, meaning that it is highly consistent with the degree of urban economic development. The majority of the counties and cities in the northwest of the study area have a strong economic foundation and better accessibility, which can provide good capital, talent, market, and other necessary conditions for the development of ecotourism and its related industries. In contrast, the cities in the southeast of the country have a slower rate of economic development, fewer public service facilities for tourists, and fewer core attraction elements for ecotourism, which has a lot of room for improvement.

## 5.2 Features of the Ili River Valley's ecotourism development level's spatiotemporal evolution

**5.2.1 Evolutionary trajectory of the center of gravity.** The center of gravity deviation trajectory map and the standard deviation ellipse of the Ili River Valley's urban ecotourism development level for each year between 2010 and 2019 were created using the ArcGIS 10.8 spatial statistical analysis tool (Fig. 5). In essence, a steady offset pattern is created from northwest to southeast. When looking at the standard deviation ellipse for each year, the direction is "northwest-southeast," which basically encompasses most of the counties and cities in the research area. The main axis and sub-axis radius contraction and rotation angle change are also smaller, indicating that the degree of clustering of ecotourism development level in the Ili River Valley has gradually increased over the study period.

Three stages can be distinguished in the center of gravity shift process: the center of gravity shifted to the southeast between 2010 and 2014; it shifted gradually to the northeast between 2015 and 2017; and it shifted to the northwest again, this time more significantly, between

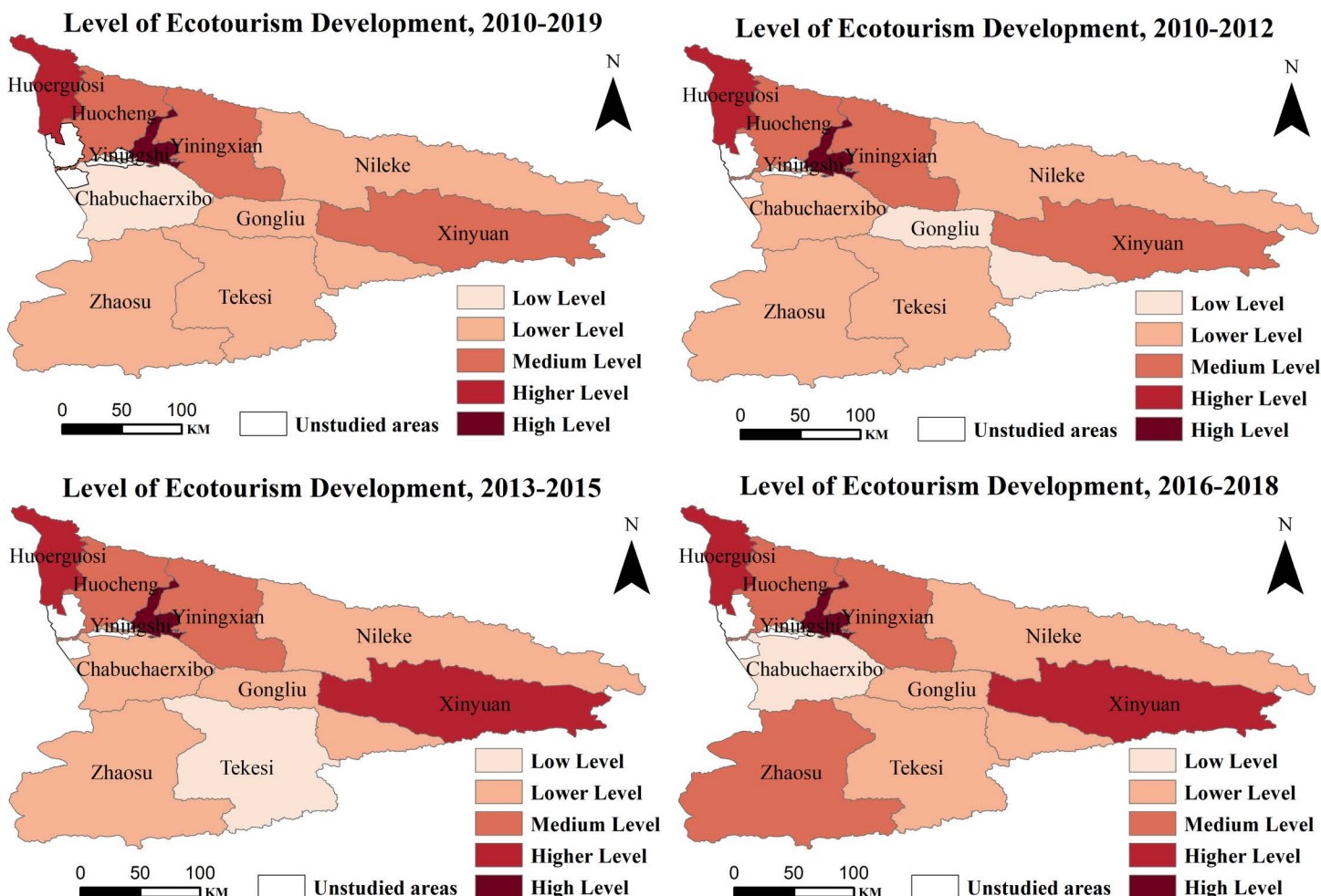

**Fig 4. The Ili River Valley's degree of ecotourism development throughout time and space, 2010–2019.** This map is based on the standard map with review number Xin S(2022)021 downloaded from the geographic information service platform of the Xinjiang Uygur Autonomous Region Department of Natural Resources, with no modifications to the base map(Same as below).

2018 and 2019. In general, the center of gravity shift lies in the study area's central west. This is due to the fact that the northwest cities' high degree of social and economic development, together with their continuous pursuit of favorable avenues for the growth of urban ecotourism, significantly raised demand for travel and raised the degree of ecotourism development. Early on, the growth of the local ecotourism business was influenced by the favorable natural conditions of the southeast counties and cities in the study area. But as time went on, the economy's slow growth was unable to offer a reliable source and assurance for the growth of ecotourism in its metropolis.

**5.2.2 Dynamic evolution characteristics.** In order to better understand the dynamic evolution law of ecotourism development level over the study period, this work constructs the standard Markov chain and the modified spatial Markov chain to analyze its rank transfer. The Markov transfer probability matrix is computed using the five levels defined in the preceding section: low level, lower level, medium level, higher level, and high level. In this case, the diagonal numbers show the likelihood of a transfer between ranks, while the diagonal values show the likelihood of no transfer at all.

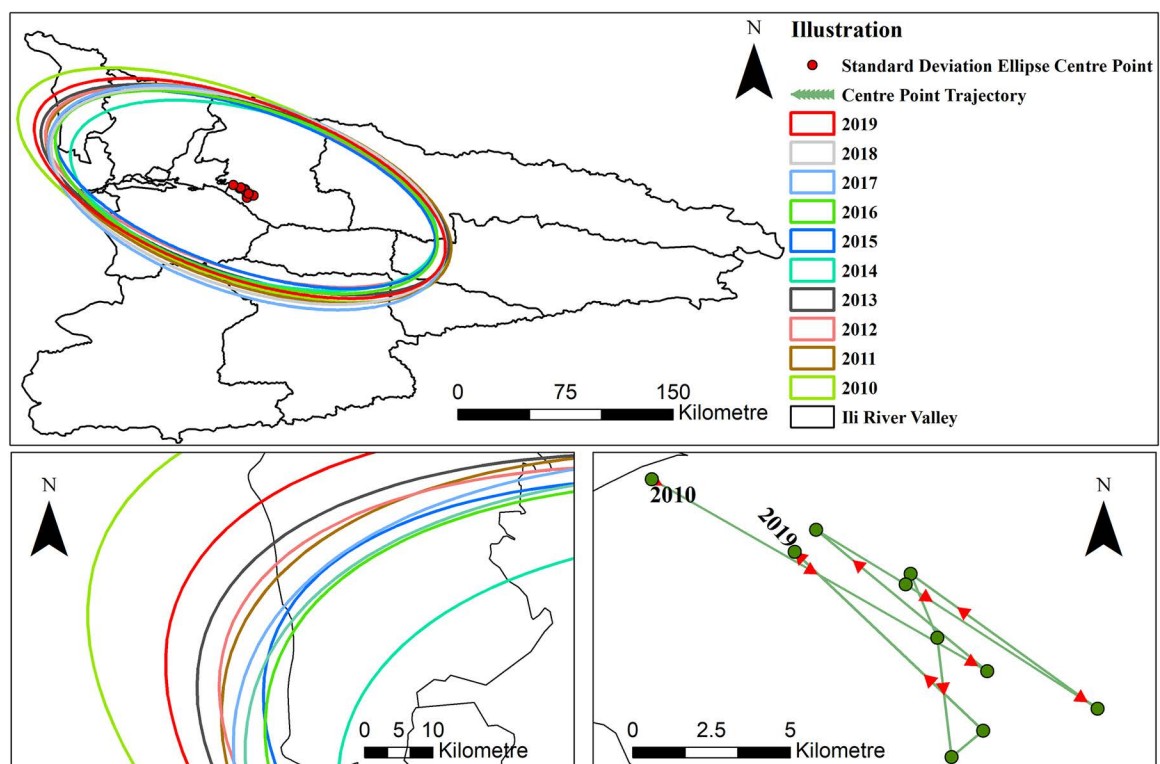

**Fig 5. Ecotourism development level standard deviation ellipse and migration trajectory of its center of gravity in the Ili River Valley.** This map is based on the standard map with review number Xin S(2022)021 downloaded from the geographic information service platform of the Xinjiang Uygur Autonomous Region Department of Natural Resources, with no modifications to the base map(Same as below).

(1) Traditional Markov chain analysis

The results show (Table 2): ① low level, higher level, high level maximum value is distributed on the diagonal, indicating that the three levels of counties and cities in subsequent years still maintain the initial state of the probability of greater stability; ② lower level of the diagonal value is smaller than the value of the non-diagonal, indicating that in the subsequent years are more fluctuating, the stability of the price is poor, in which the probability of 47.18% of the downward transfer, 17.65% of the probability of upward transfer, and 11.77% probability of leapfrog transfer; ③ medium-level counties and cities to the upper and lower neighboring areas of the transfer of the probability of roughly equal, there is no obvious upward or downward trend.

(2) Spatial Markov chain analysis

There is a geographical spillover impact on the evolution of the ecotourism development level in the Ili River Valley if the transfer of the ecotourism development level alters when the domain backdrop is taken into consideration. To determine if the spatial lag effect is statistically significant, the hypothesis test must be performed, which makes the assumption that the transfer of the type of ecotourism development level is independent of both the type of surrounding state and each other. The test formula is:

$$Q_b = -2log\left\{\prod_{l=1}^{k}\prod_{i=1}^{k}\prod_{j=1}^{k}\left[\frac{m_{ij}}{m_{ij}(S)}\right]^{n_{ij}(S)}\right\} \tag{10}$$

**Table 2. Conventional Markov transfer probability matrix for the Ili River Valley's degree of ecotourism development, 2010–2019.**

| t/t⁺1 | Low level | Lower level | Medium level | Higher level | High level |
|---|---|---|---|---|---|
| Low level | 0.44444 | 0.27778 | 0.16667 | 0.05556 | 0.05556 |
| Lower level | 0.41176 | 0.29412 | 0.17647 | 0.11765 | 0 |
| Medium level | 0.22222 | 0.27778 | 0.27778 | 0.22222 | 0 |
| Higher level | 0 | 0.05263 | 0.31579 | 0.47368 | 0.15789 |
| High level | 0.05556 | 0 | 0 | 0.16667 | 0.77778 |

Where: $m$ is the conventional Markov transfer probability and $k$ is the number of ecotourism development level kinds (in this paper, $k = 5$): $Q_b$ follows the chi-square distribution with degrees of freedom of $k(k+1)^2$, $m_{ij}(S)$ represents the spatial Markov transfer probability of the nearby state type $S$, and $n_{ij}(S)$ represents the number of counties and cities that have the spatial Markov transfer of the surrounding state type $S$. The hypothesis that type shifts in the degree of ecotourism development in the Ili River Valley are geographically independent of one another is rejected when the degrees of freedom are set to 80. $Q_b = 187.53 > \chi^2(80) = 124.84$ at the confidence level of $\alpha = 0.001$.

According to Table 3 above, the spatial Markov transfer probability matrix results show that when the domain type is low level, low level counties and cities have a 50% chance of leapfrogging to an intermediate level and a 50% chance of transferring upward to a lower level. In contrast, lower level counties and cities have a 66.67% chance of transferring downward to a low level and a 33.3% chance of leapfrogging to a higher level, while higher level counties and cities have a 33.3% chance of maintaining the initial level with a 33.3% chance of shifting upward and a 66.7% chance of shifting upward. When the domain type is lower level, the probability of the low level counties and cities in the Ili River Valley ecotourism development level remaining at the original level is 62.5%, the probability of upward transfer to lower level is 25%, and the probability of leapfrog transfer to higher level is 12.5%; however, the probability of the lower level counties and cities remaining at the original level and upward transfer to medium level are both 50%; medium-level counties and cities have a 25% chance of staying at their current level and a 50% chance of shifting to a lower level; higher level counties and cities have a 60% chance of staying at their current level and a 20% chance of shifting upward and downward, respectively; and, lastly, high level counties and cities have a 50% chance of staying at their current level and a 50% chance of shifting downward to a higher level.

When the domain type is medium level, the region's low level counties and cities have a 33.3% chance of maintaining their original level, transferring upward, or leapfrogging. The likelihood of a downward transfer to the lower level is 71.43 in the lower level counties, whereas the likelihood of an upward transfer and preservation of the initial level is 14.29%. Medium-level counties and cities have a 25% chance of staying at their current level, a 50% chance of transferring downward, and a 25% chance of dropping to a low level. The likelihood of higher level counties and cities staying at their current level and dropping to a lower level is 50%; the likelihood of high level counties and cities staying at their current level is 71.43; and the likelihood of a downward transfer is 28.57%. If the domain type is at a higher level, there is a 50% chance of no transfer and a 50% chance of leapfrog transfer to a higher level under the radiation drive of nearby cities and counties. There is a 33.3% chance that the middle-level counties will rise, fall, and then rise again. Lastly, the chance of cliff transfer to the low level was 11.1%, while the high level of counties and cities stayed at the initial level of 88.89%. There is a 42.86 percent chance that middle-level counties and cities in this region will stay at their current level and move upward when the domain type is high level, and a 14.27% chance that they will go downhill to the lower level. There is a 55.56% chance that the

**Table 3. The Ili River Valley's ecotourism development level from 2010 to 2019 is represented via a spatial Markov transfer probability matrix.**

| Type of field | t/t⁺1 | Low level | Lower level | Medium level | Higher level | High level |
|---|---|---|---|---|---|---|
| Low level | 1 | 0 | 0.5 | 0.5 | 0 | 0 |
| | 2 | 0.66667 | 0 | 0 | 0.33333 | 0 |
| | 3 | 0 | 0 | 0 | 0 | 0 |
| | 4 | 0 | 0 | 0 | 0.33333 | 0.66667 |
| | 5 | 0 | 0 | 0 | 0 | 0 |
| Lower level | 1 | 0.625 | 0.25 | 0 | 0.125 | 0 |
| | 2 | 0 | 0.5 | 0.5 | 0 | 0 |
| | 3 | 0.5 | 0.25 | 0.25 | 0 | 0 |
| | 4 | 0 | 0 | 0.2 | 0.6 | 0.2 |
| | 5 | 0 | 0 | 0 | 0.5 | 0.5 |
| Medium level | 1 | 0.33333 | 0.33333 | 0.33333 | 0 | 0 |
| | 2 | 0.71429 | 0.14286 | 0.14286 | 0 | 0 |
| | 3 | 0.25 | 0.5 | 0.25 | 0 | 0 |
| | 4 | 0 | 0.5 | 0 | 0.5 | 0 |
| | 5 | 0 | 0 | 0 | 0.28571 | 0.71429 |
| Higher level | 1 | 0.5 | 0 | 0 | 0 | 0.5 |
| | 2 | 0 | 1 | 0 | 0 | 0 |
| | 3 | 0.33333 | 0.33333 | 0 | 0.33333 | 0 |
| | 4 | 0 | 0 | 0 | 0 | 0 |
| | 5 | 0.11111 | 0 | 0 | 0 | 0.88889 |
| High level | 1 | 0 | 0 | 0 | 0 | 0 |
| | 2 | 0 | 0 | 0 | 1 | 0 |
| | 3 | 0 | 0.14286 | 0.42857 | 0.42857 | 0 |
| | 4 | 0 | 0 | 0.55556 | 0.44444 | 0 |
| | 5 | 0 | 0 | 0 | 0 | 0 |

upper level of counties and cities will drop to the medium level, while 44.44% of them will stay at the initial level.

Spatial factors significantly affect the dynamic transfer of ecotourism development level in the Ili River Valley, according to the analysis of spatial Markov chain data. Generally speaking, a low neighborhood level will impede regional development and even lower the degree of ecotourism development; a high neighborhood level contributes positively to regional development and can lessen the likelihood of its downward transfer. This could be as a result of the wealthy bordering counties and cities having a wealth of financial, human, and technical resources, among other benefits, which can help to support the region's ecotourism industry's sustainable growth.

## 5.3 Examination of the factors affecting the Ili river valley's degree of ecotourism development

**5.3.1 Selection of influencing factors.** With reference to prior research [53–55], the index system was created by choosing factors from six aspects: economic development level, urbanization level, tourism reception capacity, tourism income level, ecological environment level, and industrial structure level. This allowed for an exploration of the factors influencing the development level of ecotourism in the Ili River Valley (Table 4).①Economic development level. Regional economic development serves as the foundation for tourism growth, and

Table 4. Indicator system of factors influencing the level of ecotourism development.

| Impact factors | Representative Indicators | Properties |
|---|---|---|
| Economic Development Level | Night Lights(X1) | Forward |
| Urbanization level | Urbanization rate(X2) | Forward |
| Tourism Reception Capacity | Number of Tourism Reception(X3) | Forward |
| Tourism Income Level | Gross Tourism Income(X4) | Forward |
| Ecological Environment Level | NDVI(X5) | Forward |
| Industrial Structure Level | Tertiary Industry Output as % of GDP(X6) | Forward |

the night lights data is used to show the degree of economic development.② Urbanization level. The higher the level of urbanization, the better the regional infrastructure, and the higher the level of public services. The urbanization rate is used to represent the level of urbanization.③Tourism Reception Capacity. Tourism reception ability is the basis of tourism development. Only when the tourism reception capacity reaches a certain level can it meet the needs of tourists and provide high-quality tourism experience so as to attract more tourists. The number of star-rated hotel rooms indicates the tourism reception capacity.④Tourism income level. Tourism income is the driving force of tourism development. As tourism revenue rises, destinations can allocate more finances to support infrastructure development, product innovation, and service enhancement.⑤Ecological environment level. Ecological environment is an important prerequisite for the sustainable development of ecotourism, and NDVI is selected to represent the level of ecological environment.⑥Industrial structure level. Upgrading the industrial structure can give tourism new growth impetus, and the degree of industrial development is measured by the tertiary industry's share of GDP output value.

**5.2.2 Analysis of factor detection results.** To determine the q value of the explanatory capacity of each influencing factor, geodetectors were used to factor detect the degree of ecotourism development in the Ili River Valley in 2010, 2014, 2017, and 2019, respectively. According to the findings (Table 5), each contributing factor's explanatory power on the degree of urban ecotourism growth is in the following order of magnitude: tourism income level (X4)> economic development level (X1)> tourism reception capacity (X3)> industrial structure level (X6)> urbanization level (X2)> ecological environment level (X5).

The impact of both economic development (X1) and tourism income (X4) on the development of ecotourism has increased, with both factors now considered dominant and significant. During the initial phase of the study, ecotourism is economically reliant on favorable tourism income and economic development conditions. In the later stages of the study, ecotourism development and the level of urbanization (X2) mutually promote and depend on each other. The demand for significant income generated by the tourism industry plays a more significant role in promoting urbanization. The explanatory power of the industrial development level (X6) is slightly reduced, indicating that the optimization of industrial structure has weakened its role in promoting ecotourism development. The tertiary industry provides strong support for tourism and related industries and still plays an irreplaceable role in ecotourism. The strength of the influence of tourism reception capacity (X3) gradually increases, with an average explanatory power of over 60%. This indicates that improving tourism reception capacity effectively stimulates an increase in tourism supply. In contrast, the level of the ecological environment (X5) has a smaller degree of influence, but its explanatory power shows a fluctuating upward trend. This suggests that the ecological environment, which serves as the basis for ecotourism's ability to yield ecological advantages, continues to contribute favorably to the growth of ecotourism.

**Table 5. Findings from the identification of factors affecting the Ili River Valley's degree of ecotourism development in 2010, 2014, 2017, and 2019.**

| Influences | 2010 | 2014 | 2017 | 2019 | Average | Ranking |
|---|---|---|---|---|---|---|
| X1 | 0.4453** | 0.6** | 0.8011* | 0.8167** | 0.6658 | 2 |
| X2 | 0.3438** | 0.1111 | 0.697* | 0.6875** | 0.4598 | 5 |
| X3 | 0.3945** | 0.8095*** | 0.6916** | 0.6875** | 0.6458 | 3 |
| X4 | 0.8958*** | 0.6** | 0.8011*** | 0.8958*** | 0.7982 | 1 |
| X5 | 0.1406* | 0.2593** | 0.1023 | 0.5667** | 0.2672 | 6 |
| X6 | 0.7917** | 0.1407* | 0.6591** | 0.7667** | 0.5898 | 4 |

**Note:** Significant levels of 0.1, 0.05, and 0.01 are denoted by the symbols *, **, and ***, respectively.

**5.2.3 Analysis of Interaction Detection Results.** Upon further analysis of the interaction between the two influencing factors (as shown in Fig 6), it is evident that the explanatory power of each factor is higher after interaction than during single-factor action. This illustrates how the interaction relationship is enhanced by two factors, suggesting that the growth of ecotourism in the Ili River Valley is the consequence of several forces working together. Notably, there is no waning or loss of mutual independence among these elements, and their cooperation has a greater impact on the growth of ecotourism.

The level of tourism income and the level of economic development, along with other factors, interacted in a very significant way in 2010. This revealed that ecotourism's growth at this time was clearly dependent on economic factors and that these factors became the main drivers of ecotourism's growth. When the ecological environment and other factors interacted in 2014, the explanatory power of the interaction between the tourism reception capacity and the other factors was significant and showed varying degrees of two-factor enhancement. The explanatory power of the interaction was over 80%. The relationship between the degree of economic development and other factors was particularly significant in 2017, and the level of tourism income and the level of industrial structure interacting with other factors continued to have a high influence. In 2017, the balance of the interaction strength between the factors increased. At this point, the degree of tourism income, economic development, and industrial structure interact with other elements to primarily influence the growth of ecotourism. The significance of the relationship between the ecological environment and other factors has grown in 2019, and the explanatory power of all of them is over 77%. Additionally, the strength of the relationship between the level of economic development and other factors has somewhat improved, suggesting that as economic development increases, ecotourism development progressively improves environmental requirements.

## 6. Conclusion and Discussion

### 6.1. Conclusion

In this study, the degree of ecotourism development in the Ili River Valley's counties and cities is measured from 2010 to 2019, and its spatiotemporal evolution characteristics and affecting variables are examined. The primary findings are:

(1) At the level of each dimension, the development level of each dimension showed an increasing and then decreasing trend, with the ecotourism environment showing the highest volatility, the ecotourism site social sustainability and sustainable development potential coming second, and the ecotourism economy showing the lowest volatility in terms of sustainable development. While the development index for the tourist economy shows a regional distribution that is "high in the northeast and low in the southwest," the

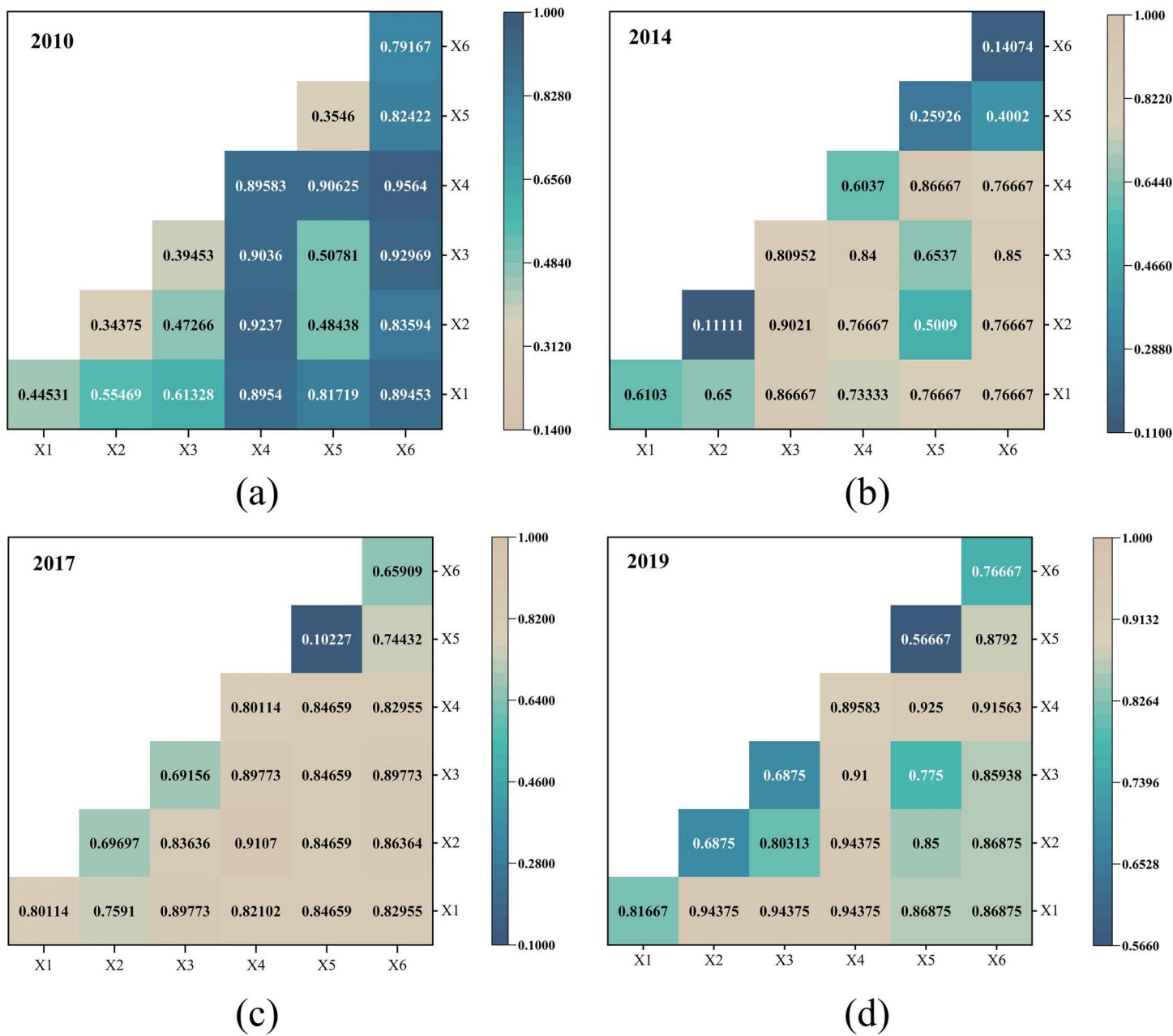

**Fig 6. Findings from the interaction detection of variables affecting the Ili River Valley's degree of ecotourism development in 2010, 2014, 2017, and 2019.**

development index for the ecotourism environment shows a spatial distribution that is "high in the west and low in the east." Its spatial distribution is "high in the north and low in the south," indicating a clear north-south division in the ecotourism community. The potential for sustainable development of ecotourism shows a spatial distribution pattern with the center of the country as a depression and a gradual increase in the outward direction.

(2) The degree of comprehensive development has been steadily increasing, with notable gradient disparities in spatial distribution that essentially form a dynamic evolutionary spatial

pattern of "high in the north and low in the south." There is a tendency for lower-level cities to contract in the southwest and for lower-level counties and cities to diffuse in the southeast, while high-level and higher-level cities are concentrated in the northwest and medium-level counties and cities surround higher-level cities.

(3) The "northwest-southeast" direction at the level of spatial and temporal evolution is indicated by the standard deviation ellipse for each year, which essentially forms a stable offset law from northwest to southeast. Furthermore, the dynamic transfer of the degree of ecotourism growth in the Ili River Valley is significantly influenced by spatial considerations. High levels of neighbors have a positive pulling impact on the region's development and can lessen the likelihood of its downward transfer, but low levels will impede the region's growth and even lower the level of ecotourism development.

(4) As for the driving factors, the growth of ecotourism is significantly influenced by the level of economic development and tourism income, while the impact of tourism reception capacity and industrial structure is steadily growing. The promotion effect of ecological environment level is not significant; the two factors' interaction effect is stronger than the force of either factor alone, demonstrating a two-factor enhancement of the interaction relationship.

## 6.2 Discussion

This paper expands the scope of ecotourism research and methodological framework by discussing the growth of ecotourism in the Ili River Valley. It is an important study of the sustainable development of tourism and the change in the development mode, and it has some reference value for the expansion of ecotourism in the Ili River Valley. This essay supports ecotourism's beneficial contribution to the growth of sustainable tourism. It is necessary to do practical research to ascertain whether ecotourism can effectively address the issue of sustainable tourism development. It's also critical to investigate whether ecotourism will put further strain on the natural environment of popular tourist attractions and how it may be developed in river valley regions. More research is necessary to answer these questions.

The Ili River Valley is a highly active region in the tourism economy of Northwest China, boasting abundant natural and cultural tourism resources. Its inherent advantage lies in the development of ecotourism. According to research findings, ecotourism in the Ili River Valley is not growing quickly and is dispersed unevenly. In light of this, the following recommendations are made in this article for the superior growth of ecotourism in the Ili River Valley.

First and foremost, certain steps should be made to raise the overall level of development and fully utilize the function of policy guidance. While integrating ecotourism resources, optimizing resource allocation, and fostering regional tourism cooperation are important aspects of the "Belt and Road" development, tourism stakeholders should also fully understand the significance of ecotourism for the sustainable growth of the industry and use ecological civilization as a guide to improve the quantity and quality of ecotourism environments. Secondly, try to address the issue of uneven development and lessen regional development disparities in accordance with local realities. While the low-level counties and cities, represented by Chabuchaerxibo Autonomous County and Tekesi County, should increase industrial assistance, thoroughly explore ecological resources, and rely on the radiation demonstration of the high-level counties and cities to push forward the high-quality development of the economy, the core area, represented by Yining City and Huoerguosi City, should fully utilize the advantages of capital and technology and promote the coordinated development of economic development and ecological protection. Lastly, to compensate for the deficiencies, strengthen the

dominant factors' driving role, and encourage the growth of ecotourism power. To effectively meet the growing demand for eco-tourism in cities, maximize the benefits of economic development, boost ecological and environmental protection inputs, and create a strong foundation for eco-tourism development. You should also increase the rate of urbanization and keep increasing the supply of eco-tourism products.

However, there are still certain restrictions on this article. To begin with, this study's evaluation index system needs more work because it is constrained by the availability of the data. Secondly, future studies can consider using ecological resources interest point data, questionnaire data, and other sources to enhance the evaluation index system. Optimizing the assessment index system for the degree of ecotourism development is the goal of the paper. The study focuses on six major factors that affect ecotourism development. However, it does not consider the impact of natural conditions, policy conditions, and other factors. Future research will explore the impact of various factors on ecotourism comprehensively. This paper focuses on the valley as the research scale. The study's conclusions regarding the interpretation of small-scale ecotourism have some limitations. Future research should explore ecotourism phenomena at different scales to further enrich the practical and promotional value of the study.

## Author contributions

**Investigation:** Xinyu Zhao.

**Supervision:** Haojie Sun.

**Validation:** Xueting Xu.

**Writing – original draft:** Pengkai Zhao.

**Writing – review & editing:** Jiangling Hu, Changying Song.

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
