## [Decision Letter · Decision Letter 0]

27 Sep 2024

PONE-D-24-13106Spatial and Temporal Evolution of Ecotourism Development Level and Its Driving Factors under the Perspective of Sustainable Development:The Case of Ili River ValleyPLOS ONE

Dear Dr. Zhao,

Thank you for submitting your manuscript to PLOS ONE. After careful consideration, we feel that it has merit but does not fully meet PLOS ONE’s publication criteria as it currently stands. Therefore, we invite you to submit a revised version of the manuscript that addresses the points raised during the review process.

We look forward to receiving your revised manuscript.

Kind regards,

Muzafar Riyaz, Ph.D.

Academic Editor

PLOS ONE

Journal Requirements:

2. Thank you for submitting the above manuscript to PLOS ONE. During our internal evaluation of the manuscript, we found significant text overlap between your submission and previous work in the [introduction, conclusion, etc.].

Please revise the manuscript to rephrase the duplicated text, cite your sources, and provide details as to how the current manuscript advances on previous work. Please note that further consideration is dependent on the submission of a manuscript that addresses these concerns about the overlap in text with published work.

[If the overlap is with the authors’ own works: Moreover, upon submission, authors must confirm that the manuscript, or any related manuscript, is not currently under consideration or accepted elsewhere. If related work has been submitted to PLOS ONE or elsewhere, authors must include a copy with the submitted article. Reviewers will be asked to comment on the overlap between related submissions (http://journals.plos.org/plosone/s/submission-guidelines#loc-related-manuscripts).]

We will carefully review your manuscript upon resubmission and further consideration of the manuscript is dependent on the text overlap being addressed in full. Please ensure that your revision is thorough as failure to address the concerns to our satisfaction may result in your submission not being considered further.

 [This research was supported by the Xinjiang Uygur Autonomous Region Social Science Foundation Project Self-help (Project No. 2023BYJ033).]. 

5. We note that [Figures 1,3,4 and 5] in your submission contain [map/satellite] images which may be copyrighted. All PLOS content is published under the Creative Commons Attribution License (CC BY 4.0), which means that the manuscript, images, and Supporting Information files will be freely available online, and any third party is permitted to access, download, copy, distribute, and use these materials in any way, even commercially, with proper attribution. For these reasons, we cannot publish previously copyrighted maps or satellite images created using proprietary data, such as Google software (Google Maps, Street View, and Earth). For more information, see our copyright guidelines: http://journals.plos.org/plosone/s/licenses-and-copyright.

We require you to either (1) present written permission from the copyright holder to publish these figures specifically under the CC BY 4.0 license, or (2) remove the figures from your submission.

a. You may seek permission from the original copyright holder of Figures 1,3,4 and 5 to publish the content specifically under the CC BY 4.0 license. 

Reviewers' comments:

Reviewer's Responses to Questions

**Comments to the Author**

1. Is the manuscript technically sound, and do the data support the conclusions?

Reviewer #1: Yes

2. Has the statistical analysis been performed appropriately and rigorously? 

Reviewer #1: Yes

3. Have the authors made all data underlying the findings in their manuscript fully available?

Reviewer #1: Yes

4. Is the manuscript presented in an intelligible fashion and written in standard English?

Reviewer #1: Yes

5. Review Comments to the Author

Reviewer #1: This paper constructs an indicator system with a focus on eco-tourism, exploring the impact of eco-tourism on the economic development of the Ili River Valley while providing a theoretical basis. The topic is innovative and holds significant research value, with a clear logical approach in both the analysis methods and the construction of indicators.

However, there are still some issues. The author should revise the paper according to the following suggestions:

1.Introduction Section: The specific challenges faced by eco-tourism in the Ili River Valley and the importance of its development should be described in greater detail. This not only includes the pressures of environmental protection and sustainable development but also addresses how to reasonably develop tourism resources while preserving natural ecosystems, thereby achieving a win-win situation for both ecological and economic benefits. Supplementing these aspects will help deepen readers' understanding of the research background and motivation, while also providing a more solid theoretical foundation for subsequent research.

2.Results Section: The figures and tables could be added or improved to enhance their visual appeal and effectiveness in conveying information. Specifically, adjusting the color scheme could make the different parts of the figures more distinct, thereby improving the readability of the information and the overall aesthetic quality of the figures. Additionally, for figures involving geographic information, considering adjustments to the map projection coordinate systems could more accurately reflect the geographical characteristics of the study area, making the map presentations more consistent with actual conditions.

3.References Section: Some of the references cited are from earlier years; it is necessary to reduce the number of older references and increase the inclusion of literature published within the last five years.

4.Grammar and Sentence Structure: There are sloppy grammatical and editorial issues everywhere in this paper. Many of them could cause difficulties to the readers

6. PLOS authors have the option to publish the peer review history of their article (what does this mean? ). If published, this will include your full peer review and any attached files.

**Do you want your identity to be public for this peer review?** For information about this choice, including consent withdrawal, please see our Privacy Policy .

Reviewer #1: No

---

## [Author Response · Author response to Decision Letter 1]

30 Oct 2024

Responds to the reviewers' comments:

(1)Introduction Section: The specific challenges faced by eco-tourism in the Ili River Valley and the importance of its development should be described in greater detail. This not only includes the pressures of environmental protection and sustainable development but also addresses how to reasonably develop tourism resources while preserving natural ecosystems, thereby achieving a win-win situation for both ecological and economic benefits. Supplementing these aspects will help deepen readers' understanding of the research background and motivation, while also providing a more solid theoretical foundation for subsequent research.

Thank you for your valuable comments on the introductory section of our manuscript. We are grateful for your suggestions and have taken the following steps to address them: We have expanded the Introduction section to describe in more detail the specific challenges facing ecotourism in the Ili River Valley and the importance of its sustainable development. We believe that these additions will greatly enhance the reader's understanding of the context and motivation for the study. By providing a firmer theoretical foundation, the revised introduction will better support subsequent research in the manuscript. Relevant changes are reflected on page 4, lines 60-75 of the revised manuscript. Thank you again for your interest in our work. We look forward to your further review.

(2) Results Section: The figures and tables could be added or improved to enhance their visual appeal and effectiveness in conveying information. Specifically, adjusting the color scheme could make the different parts of the figures more distinct, thereby improving the readability of the information and the overall aesthetic quality of the figures. Additionally, for figures involving geographic information, considering adjustments to the map projection coordinate systems could more accurately reflect the geographical characteristics of the study area, making the map presentations more consistent with actual conditions.

Thank you for your valuable suggestions on the results section. We have taken the following steps to address them:

①We have carefully adjusted the color scheme of the charts to make the different sections more distinct. To do this, we have chosen colors with better contrast and visual appeal. This was done to improve the readability of the information in the charts and to improve the overall aesthetics of the charts. In addition, we reassessed and adjusted the map projection coordinate system for charts with geographic information.

②We reviewed the layout and formatting of tables to ensure that they were legible. This included adjusting column widths and font sizes and adding appropriate headings and footnotes where necessary.

We believe that these improvements will greatly enhance the visual appeal of the figures and tables and the effectiveness of the information conveyed. We look forward to your review of the revised manuscript. Thank you again for your attention and guidance.

(3) References Section: Some of the references cited are from earlier years; it is necessary to reduce the number of older references and increase the inclusion of literature published within the last five years.

Thank you for your valuable feedback on the references in our manuscripts. We appreciate your concern in this regard and understand the importance of having a balanced and up-to-date list of references.

We have conducted a thorough review of the references and have taken steps to address your concerns. In the revised version, we have carefully evaluated each reference and have tried to reduce the number of older references. We removed references that were less relevant or could be replaced with newer works. We also actively searched for and included additional literature published within the last five years. These new references enhance the currency and relevance of our research and indicate that our work references the latest research in the field. This change is primarily reflected in documents 1, 2, 7, 8, 11, 21, 33, 34, 35, and 49 in the list of references and has been revised in the text on pages 3, 45-50, page 5, 80-90, and page 10, line 171.

We look forward to your further review and consideration of our revised manuscript. Thank you again for your feedback and guidance.

(4) Grammar and Sentence Structure: There are sloppy grammatical and editorial issues everywhere in this paper. Many of them could cause difficulties to the readers.

Thank you for bringing to our attention the issues regarding grammar and sentence structure in our paper. We sincerely apologize for these shortcomings and understand the potential difficulties they may cause for readers.

We have thoroughly reviewed the manuscript and made significant efforts to correct the sloppy grammatical and editorial issues. We have carefully proofread each sentence, checking for proper grammar, punctuation, and word usage. We have also paid close attention to the overall flow and coherence of the text to ensure that it is presented in a clear and understandable manner.

We are committed to improving the quality of our work and appreciate your feedback, which has been invaluable in helping us to identify and address these areas for improvement. We believe that the revised manuscript now meets a higher standard of grammar and sentence structure and will be more accessible to readers.

Thank you again for your time and consideration. We look forward to your further review of the revised manuscript.

---

## [Editor Report · Decision Letter 1]

28 Nov 2024

Spatial and Temporal Evolution of Ecotourism Development Level and Its Driving Factors under the Perspective of Sustainable Development:The Case of Ili River Valley

PONE-D-24-13106R1

Dear Dr. Sun,

We’re pleased to inform you that your manuscript has been judged scientifically suitable for publication and will be formally accepted for publication once it meets all outstanding technical requirements.

Kind regards,

Muzafar Riyaz, Ph.D.

Academic Editor

PLOS ONE

---

## [Editor Report · Acceptance letter]

PONE-D-24-13106R1

PLOS ONE

Dear Dr. Sun,

I'm pleased to inform you that your manuscript has been deemed suitable for publication in PLOS ONE. Congratulations! Your manuscript is now being handed over to our production team.

Kind regards,

on behalf of

Dr. Muzafar Riyaz

Academic Editor

PLOS ONE